# A Bibliometric Analysis of Wearable Device Research Trends 2001–2022—A Study on the Reversal of Number of Publications and Research Trends in China and the USA

**DOI:** 10.3390/ijerph192416427

**Published:** 2022-12-07

**Authors:** Itsuki Kageyama, Karin Kurata, Shuto Miyashita, Yeongjoo Lim, Shintaro Sengoku, Kota Kodama

**Affiliations:** 1Graduate School of Technology Management, Ritsumeikan University, 2-150 Iwakuracho, Ibaraki 567-8570, Japan; 2School of Environment and Society, Tokyo Institute of Technology, Tokyo 108-0023, Japan; 3Center for Research and Education on Drug Discovery, The Graduate School of Pharmaceutical Sciences, Hokkaido University, Sapporo 060-0812, Japan

**Keywords:** bibliometric analysis, cluster analysis, co-occurrence analysis, wearable device

## Abstract

In recent years, Wearable Devices have been used in a wide variety of applications and fields, but because they span so many different disciplines, it is difficult to ascertain the intellectual structure of this entire research domain. No review encompasses the whole research domain related to Wearable Devices. In this study, we collected articles on wearable devices from 2001 to 2022 and quantitatively organized them by bibliometric analysis to clarify the intellectual structure of this research domain as a whole. The cluster analysis, co-occurrence analysis, and network centrality analysis were conducted on articles collected from the Web of Science. As a result, we identified one cluster that represents applied research and two clusters that represent basic research in this research domain. Furthermore, focusing on the top two countries contributing to this research domain, China and the USA., it was confirmed that China is extremely inclined toward basic research and the USA. toward applied research, indicating that applied and basic research are in balance. The basic intellectual structure of this cross-sectional research domain was identified. The results summarize the current state of research related to Wearable Devices and provide insight into trends.

## 1. Introduction

Wearable means “something that can be worn”. Wearable devices or wearable technology is a general term for small electronic devices, mobile devices, computers, microchips, and other information devices with wireless communication capabilities that can be worn on the human body as accessories, gadgets, or clothing [1]. In recent years, various wearable devices have been developed, including wristwatch-type devices such as the Apple Watch, Fitbit, Samsung Gear, Microsoft Band, and Garmin, and eyeglass-type, ring-type, shoe-type, and clothing-type devices. According to a report by Mordor Intelligence [2], global shipments of smart wearables are expected to reach 266.3 million units by 2020 and 776.23 million units by 2026. International Data Corporation [3] expects full-year shipments to remain flat at 535.5 million units in 2022 due to rising global inflation, semiconductor shortages, and concerns over the recession caused by the global situation but expects growth to pick up again in 2023. According to Reportlinker.com [4], the wearable technology market is expected to grow to $61.32 billion between 2022 and 2026 at a CAGR of 14.31%. Wearable devices are used in a wide variety of settings, including education, medicine, entertainment, gaming, sports, and music; based on Vandrico’s wearable device database, there were 431 registered devices as of 21 September 2022 [5]. The top registered categories were lifestyle (225 devices), fitness (188 devices), and medicine (88 devices), indicating a strong interest in health management (it should be noted, however, that the Vandrico database has some devices classified into multiple categories). The most common health use of wearable devices is to monitor physiological and biochemical parameters of daily life and information associated with human behaviour. Heart rate information obtained from electrocardiograms and phonocardiograms, blood pressure, blood oxygen saturation, body temperature, and physical activity are commonly used. Traditionally, these parameters have been used to monitor physical information such as sleep [6,7] and exercise [8,9,10]. In recent years, based on these parameters, there is also a focus on developing devices, algorithms, and systems for monitoring the body’s stress state (physical and psychological stress) [11,12,13]. In particular, many efforts have been made to monitor the biometric and physical information of workers in the work environment [14,15,16,17,18,19], to analogize workload and physical and psychological stress, and to utilize this information for labour management, work environment safety risk alerts and accident prevention [20,21,22,23,24,25,26,27,28,29].

There have been relatively few studies that have focused on comprehensive knowledge structures despite the enormous growth in wearable device research in recent years. Previous studies have reported use- or industry-focused reviews, such as a bibliometric analysis of wearable technology research in the healthcare field [30] and the application of wearable devices in safety management in the construction industry [31]. Alternatively, a bibliometric analysis of wearable device research has been reported using wearable technology and other related products (e.g., smartwatches and smart glasses) as keywords [32]. However, the use of specific keywords narrows the research domain of interest and, trend analyses are limited to qualitative reviews of research papers under particular conditions, and none of them covers the wearable device research as whole. The reason for this is thought to be that this research domain covers many different fields, making it difficult to see the research domain as a whole. Thus, previous studies intentionally narrow the target research domain by selecting the in-tended use and industry or assigning specific keywords. Therefore, this study does cover the entire wearable device. This study collects information in the research domain of wearable devices from 2001 to 2022, organizes it quantitatively through bibliometric analysis, and clarifies the intellectual structure of this industry to solve this research gap. The objective is to comprehensively and objectively clarify the entire research domain surrounding wearable devices by not limiting the industry in which wearable devices are introduced or by carefully selecting keywords in the literature collection, as in previous studies.

## 2. Methods

### 2.1. Bibliometric Analysis

Quantitative analysis, frequency distribution, and statistics of publications are an important factor in science policy in many countries and are referred to as bibliometric analysis. Bibliometric analysis is a research method that utilises mathematics, statistics, and bibliography to quantitatively analyse academic literature [33,34,35]. It is widely used to identify relationships among authors, periods, research directions, and other variables in order to identify knowledge structures, emerging trends, and tendencies in a given research domain [36]. By rigorously understanding large amounts of unstructured data, the nuances of the accumulation and evolution of scientific knowledge in established fields can be deciphered and mapped so that researchers can recognise “hidden patterns” and use them to inform decision-making processes, predict new technologies [37]. Today, bibliometric software such as VOSviewer [38,39] and scientific databases such as Web of Science have advanced and become more accessible, which is one reason why bibliometric analysis has become popular.

### 2.2. Cluster Analysis

This study used VOSviwer software, which provides visualisation of bibliometric networks, to create a co-occurrence word network and cluster analysis [39] for identifying words with similar characteristics in the collected literature and trends in research activities related to wearable devices. The results of the bibliometric analysis are often presented as visual maps using visualisation tools such as VOSviwer [39], CiteSpace [40], and HistCite [41,42]. Each visualisation tool was differentiated based on its functionality. In this study, VOSviewer was used because it specialises in visualising bibliometric networks, is suitable for visualising large networks, and has sufficient functionality to extract cluster information.

VOSviewer arranges the nodes in a network in a two-dimensional space, assigns them to exactly one cluster, and visualises them according to their individual colours [43]. A cluster comprises a set of closely related nodes. The size of a circle represents its importance/frequency within the bibliographic network (what exactly a circle means depends on the object of analysis), and the closer the distance between circles, the stronger the bibliographic relevance. The VOSviewer uses a smart local moving algorithm [44].

### 2.3. Co-Occurrence Analysis of Keywords

The (author) keywords provided to a paper reflect the core content of the research paper. Keyword co-occurrence analysis is an important quantitative method in bibliometric analysis for investigating scientific constructs based on the assumption that keywords provide a consistent explanation for the content of an article [45]. This method implies that if keywords co-occur frequently in the literature, the underlying concepts and theories are closely related [46]. This study did not take steps to merge singular and plural keywords (e.g., singular “devices”, “sensors” and plural “devices”, “sensors”, etc.). In this study, what each cluster means was judged by keywords except above one. The reason is that these keywords, which are often listed by authors in wearable device research field, can generate false co-occurrence even if there is not actual relationship. Therefore, these keywords (i.e., “devices” and “devices”) exist as separate nodes and are each classified into a cluster. The association between two keywords indicates the strength of the link between them, which is numerically expressed as a numerical value, with a higher number indicating a stronger association. If a keyword has many co-occurrences, it indicates that it is central, with many connections to other words. These can be quantitatively indicated using a centrality measure [47]. Centrality is an indicator that attempts to capture from a network perspective the presence that influences others (other nodes) in a given network, the central presence, and can quantitatively express the structural characteristics of the network as a whole. The concept of network centrality emerged in the 1940s, and many centrality measures have since been proposed [48]. In this study, we used the following three centrality measures commonly used in network analysis.
Degree Centrality

The measure is expressed as the number of lines (edges) connecting a node. This indicates the number of edges connecting nodes. A node with a high degree centrality has many connections with other nodes and can be evaluated as a central node.
2.Betweenness Centrality

The measure of the degree to which a node is the shortest path to another node [49]. This indicates the extent to which a node is included in the shortest path between other nodes or relays a node. In a network of connected information, connectivity is maintained by the existence of a node that acts as a bridge between networks. If a bridging node does not exist, a situation may arise in which the information transmission path of the network senses would have collapsed. Therefore, we focused on the linkage relationship between nodes and evaluated more nodes and the central node that bridges the network.
3.Closeness centrality

Closeness centrality is an indicator of the distance between a node and all other nodes [50,51]. It takes the reciprocal of the average of the shortest path lengths between nodes and can evaluate nodes that are most closely related to other nodes.

The following settings were used in VOSviewer to perform co-occurrence analysis:
Type of analysis: Co-occurrenceUnit of analysis: All keywords

As a result, a total of 49,720 keywords existed in all target articles, and we conducted a co-occurrence analysis using the top 1000 keywords out of 4756 that exceeded the co-occurrence threshold at least five times.

### 2.4. Data Collection

The Web of Science (WoS) database was used to collect bibliographic information. WoS has strict selection criteria and carefully selects high-quality international journals based on their impact factors. WoS was selected because it has the policy of listing all items in the literature, which makes it possible to handle exhaustive bibliographic information and exclude poor-quality literature. The keyword “Wearable Device” (hereafter WD) was used to collect literature, and all relevant literature was included in the survey. The subject fields were not included. This is because analysing research fields related to wearable devices is the subject of only one survey. However, we excluded proceedings papers and books and limited the text type to “Article” to include only articles published in specialised journals. The study period was from 2001 to 2022 (31 May 2022). The final string used in the search was as follows. (ALL = (Wearable Device)) AND (DT == (“ARTICLE”)) AND (DOP == (2000-01-01/2022-05-31)). The parameters used to annotate the literature are summarised below. As a result, a final total of 20,581 publications were extracted (Figure 1). In this study, screening requirements were loosely set to cover a broad range of studies in the wearable device research domain (time span, Language, Document types). The extracted publications were exported as text files using the WoS export function, and “Full Record and Cited References” was selected as the record content to obtain all relevant information for all 43 references (Section A.1).

### 2.5. The Software Used for the Analysis

Bibliographic information obtained from the WoS was used to form bibliographic networks and clustering using VOSviewer. The network information generated by VOSviewer was analysed for centrality using MATLAB (R20022a). The results were analysed using Microsoft Excel (2019).

## 3. Results

### 3.1. Characteristics of Research Domains

Figure 2 shows the number of publications related to “Wearable Device” collected from the Web of Science database by year.

In 2001, only 22 papers were available. However, since 2014, the number of relevant papers has increased dramatically, with 4608 papers published in 2021, an increase in approximately 209 times compared to 2001. A total of 18,553 relevant papers were published between 2001 and 2021. From the results of year 2001 through 2021, a fourth polynomial regression analysis was performed to estimate the number of publications to the year of 2025 ($y=0.0032x^{4}+1.5732x^{3}-31.788x^{2}+178.15x-202.35$, $R^{2}=0.9952$) [36]. As a result, the number of papers related to the domain of research on wearable devices is expected to exceed 10,000 in 2025.

Figure 3 compares the number of papers on wearable devices by country. During the period 2001–2022, People’s Republic of China (hereafter referred to as “China”) has by far the largest number of papers, with a total of 7490 papers published, corresponding to about 37% of the total number of papers. The next largest countries were the United States (4508 papers, or about 23% of the total) and South Korea (2648 papers, or 13% of the total), indicating that China and the United States are leading in this research domain.

Figure 4 shows the co-authorship network for the ten countries that contributed the most to the literature in this research domain. All the top 10 countries belonged to a co-authorship network, indicating world-class collaboration. The co-authorship networks in the top 10 countries formed two clusters. Cluster 1 (red) was a group formed by Italy (rank 4), England (rank 5), Australia (rank 7), and Canada (rank 10), including China (rank 1) and the USA (rank 2). The group included Europe, the U.S., and China. Cluster 2 (green) comprised South Korea (rank 3), Japan (rank 6), India (rank 8), and Taiwan (rank 9), a group formed in Asian countries excluding China. In the field of wearable device research, it is clear that there is a strong and weak collaboration among regions.

A comparison of the number of papers related to wearable devices by the journal is shown in Figure 5. The number of papers in the period 2001–2022 is the largest for *Sensors* (886 papers, or approximately 4.4% of the total number of papers), followed by *ACS Applied Materials and Interfaces* (823 papers, or approximately 4% of the total number of papers). This was followed by *Nano Energy* (460 papers, corresponding to 2.3% of the total). The top 10 journal categories covered a wide range of fields, including chemistry, materials science, physics, energy, nanotechnology, electrical engineering, and information technology.

### 3.2. Co-Occurrence Analysis of Keywords

For all articles in this research domain extracted from the Web of Science between 2001 and 2022, a co-occurrence analysis was performed on all author keywords in the literature using VOSviewer to generate a co-occurrence network. The generated networks formed three custers (Figure 6 and Table 1). The top ten keywords for each cluster are shown. Each cluster had the following characteristics. Cluster 1 (red) contains the names of the wearable device research domains themselves, such as “wearable device(s)” and “wearable sensors”. Others included words related to health care such as “health”, “behaviour”, “walking”, “physical activity”, “rehabilitation”, and “disease”, as well as “biomedical monitoring (monitoring)”, “internet”, “system(s)”, and other technical words, and “machine learning”, which has been the focus of much attention in recent years. Thus, we can say that cluster 1 covers words related to applied research, such as medicine, bionics, (medical) engineering, and human–computer interaction. Cluster 2 (green) consists of words related to sensors and elements, such as “sensor”, “wearable electronics”, “carbon nanotube(s)”, “composites”, “conductivity”, and “strain sensor.” Thus, it can be said that cluster 2 is a cluster covering words related to chemistry and (electrical) engineering. Cluster 3 (blue) consists of words related to nanotechnology and compounds such as “graphene”, “oxide”, “reduced graphene oxide”, “nanowires”, “supercapacitor”, “nanostructures”, “nanosheets”, and “polypyrrole”. Cluster 3 comprised words related to nanotechnology and compounds. Thus, it can be said that cluster 3 is composed of words related to materials science (engineering). Cluster 1 covers applied research, and clusters 2 and 3 cover basic research. Of the total 1000 keywords comprising the co-occurrence network, cluster 1 accounted for about 48%, cluster 2 for about 36%, and cluster 3 for about 16%, with the applied research domain (cluster 1) and the basic research domain (clusters 2 and 3) accounting for approximately half.

### 3.3. Comparing Two Countries’ Contributions to the Research Domains (China vs. USA)

Figure 7 and Section A.2 show the number of publications by year for China and the USA in the wearable device research domain from 2001 to 2021. As of 2021, China had 1974 papers, and the USA had 866 papers, approximately twice as many as the USA. The trend in the number of papers published in both countries was divided into three periods. Period 1: The period of USA dominance. Period 1: The period from 2001 to 2013, when the number of papers published in the USA was slightly higher than that in China. During this period, China averaged 7.4 papers/year, and the USA averaged 21.6 papers/year, indicating that the USA had more papers than China, albeit slightly. Period 2: The number of papers in China and the United States competed with each other from 2014–2016. Unlike Period 1, there was almost no difference in the number of papers published in China and the USA. Period 3: The period of China’s dominance, from 2017 to 2021, when the number of papers in China reversed the trend of the number of papers in the U.S. and greatly exceeded the number of papers in the U.S. Unlike in Period 1 and Period 2, the number of papers in China and the number of papers in the USA averaged 1220/year and 686/year, respectively, and the number of papers in China greatly surpassed the number of papers in the USA. While the number of papers in the USA is becoming somewhat steady and parallel, China is still in the growth phase and is expected to increase further in the future.

We examined the centrality (i.e., importance) of both countries in their networks within the co-authorship network (Section A.3 to determine the roles of China and the USA in this research domain. The USA ranked the highest among all centrality measures. China ranked next in degree centrality and closeness centrality, but in betweenness centrality, China ranked 5th, a gap from the USA’s result. In the co-authorship network, the centrality measures of the USA and China remained the same, or the USA outranked China in some indices, indicating that the USA is central in this network, despite the difference in the number of papers. This result was also reflected in the ratio of citations per literature in both countries (citation impact) (Figure 8, Section A.2). The citation impact in both countries also did not differ significantly between China and the USA, contrary to the rapid increase in the number of publications in China in recent years. Although an effect of the calculation based on single year to capture recent increase and a time period after the appearance of a publication (often referred as citation window) should be considered [52], of course, recent trend of citation impact suggests qualitative difference of wearable device research between China and USA.

We analysed whether the author keywords used in the literature in China and the USA could be classified as clusters in the co-occurrence network (Figure 6) as of 2022 (Figure 9). In China, keywords belonging to cluster 2 predominated, followed by cluster 3, and keywords belonging to cluster 1 appeared the least (cluster 1: 7%, cluster 2: 54%, cluster 3: 39%). However, USA was dominated by keywords belonging to cluster 1, followed by cluster 2 and then cluster 3 (cluster 1: 61%, cluster 2: 28%, cluster 3: 11%).

Section A.4 shows the top 10 most-cited publications in China and the USA for the period 2001–2021. The top 10 cited publications in China were all related to materials and device research and were in the basic research domains of clusters 2 and 3 when classified according to the cluster classification.

On the other hand, in the case of the USA, in addition to basic research domains such as clusters 2 and 3, as in China, there were also some citations in the applied research domain (i.e., cluster 1) related to the utilisation of wearable devices, such as Pantelopoulos et al. [53] and Son, D. et al. [54]. The number of citations in the literature on the use of wearable devices is also presented.

## 4. Discussion

This study analysed articles in the research domain related to “Wearable Device” using bibliometric analysis. Research related to wearable devices has increased every year since 2001, with a marked increase after 2014. China and the United States were the largest contributors to wearable device research. In general, in many research domains, the country with the highest contribution is the United States. However, in the wearable device research domain, the situation is reversed, with China making the largest contribution, followed by the United States. However, when the centrality measure of the co-authorship network in the research domain was examined, it was higher in the USA than in China in terms of the number of papers, indicating that the USA constitutes the centre of the network in the research domain. In addition, the citation impact of the two countries was at the same level, contrary to the difference in the number of papers published in recent years. These results suggest two possible cases: “1. China has a low number of citations compared to the number of papers.” or “2. The USA has fewer papers than China but is often cited.”

However, comparing the cluster distribution of author keywords between China and the USA suggests the possibility that this was due to the direction of research in both countries. Comparing the cluster distribution of keywords in China and the USA, the majority of keywords in the USA are in cluster 1, while cluster 1 is extremely rare in China, indicating that the characteristics of the two countries are exactly the opposite. Cluster 1 is related to applied research, and the research domains associated with it may be diverse. This result was also reflected in the centrality analysis. Comparing betweenness centrality, China dropped in rank to 5th place compared to the USA, which ranked 1st. Betweenness centrality measures how well a country is connected to more nodes, networks, and small network groups (also called communities). In other words, the USA was shown to be connected to more communities (sectors) than China. These circumstances suggest that China’s citation impact may have been at the same level as that of the USA, contrary to the difference in the number of papers, and may reflect the different orientations toward research in the two countries.

It is also possible that the public is more interested in applied research and less interested in basic research, such as that related to clusters 2 and 3. Similar previous studies have reported that wearable-technology-related research is concentrated in domains such as health [30,55,56]. This is synonymous with the fact that research related to cluster 1 in this study is thriving but may be the result of trends or keywords cut by some countries, such as the USA. In research domains related to wearable technology, the literature related to medicine and engineering contributed the most to this research domain, with a particular focus on healthcare, including rehabilitation and disability [57]. The cluster classification in this study also confirmed both applied and basic research, and in cluster 1, keywords such as “behaviour”, “health”, and “rehabilitation” were identified as the most common, which supports these reports. The introduction of wearable devices in the health and medical fields is expected to accelerate in the future [58]. However, further technological development is essential in the wearable-device research domain, not only in terms of applications.

For example, the improved performance of lithium-ion batteries will enable longer real-world operating times and longer periods of time for information transmission or monitoring. In addition, if sensors and elements can be made smaller and lighter while maintaining the same performance, the devices themselves may become smaller and lighter, enabling operation regardless of the age or location of use. In this research domain, where both basic and applied research are important, the difference in direction between China and the USA is a desirable outcome. In other words, typically through the application in the USA, outcomes derived from the use case of wearable devices in the real world can accelerate active knowledge transfer into the research field for innovative component to improve the wearable device performances, where China has superiority, vice versa.

## 5. Conclusions

The purpose of this study was to collect literature on the entire research domain surrounding wearable devices and objectively clarify the intellectual structure of this research domain using bibliometric analysis. As a result of co-occurrence analysis of research keywords, three clusters were identified and shown to be composed of applied research domain (Cluster 1) and basic research domain (Clusters 2 and 3). In addition, China (mainly basic research domain) and the USA (mainly applied research domain) were the leaders in this research from 2001–2022. Moreover, wearable device research, which has been difficult to understand due to its extensive coverage in various fields (e.g., chemistry, energy, nanotechnology, etc.), can now be classified into two major categories: applied research and basic research. This may enable a simpler explanation of this research domain. Research on wearable devices has increased every year since 2001 and is expected to continue to grow.

The introduction of wearable devices in the health and medical fields is expected to accelerate in the future [58]. However, further technological development is essential in the wearable-device research domain, not only in terms of applications. For example, the improved performance of lithium-ion batteries will enable longer real-world operating times and longer periods of time for information transmission or monitoring. In addition, if sensors and elements can be made smaller and lighter while maintaining the same performance, the devices themselves may become smaller and lighter, enabling operation regardless of the age or location of use (e.g., recent smart watches can measure body temperature, blood flow, blood pressure, and blood oxygen levels, but research is being conducted to measure blood sugar levels [59]). Thus, technological developments and breakthroughs from basic research can create new uses (applied research), and new applied research can motivate new basic research. Additionally, it will open up new markets. In this research domain, where both fundamentals and applications are important, the different directions of China and the USA are a desirable outcome in a mutually supportive manner.

We recognize the limitations in this study. In the cluster classification, we did not unify some author keywords for simplicity because we could judge meanings of each cluster without lemmatization. To conduct more precise analysis, however, preprocessing of author keywords, such as merging singular and plural keywords or setting some stopwords will be required. Furthermore, we did not analyze trends in this research area in terms of time. This is a problem with the tool used (VOSviewer is not suitable for analyzing from a time perspective. CiteSpace would be a better choice).

The results of this study will provide comprehensive visualization data, identifying developments and trends in this research domain. Additionally, it provides improvements for further in-depth understanding. However, specific technological developments and breakthroughs may change the research trends rapidly. It should be noted that the results of this study are current information.

## 6. Limitations

This study was limited to articles extracted from the Web of Science database using the keyword “Wearable Device”. Therefore, the results of this study are limited to relevant data. Thus, similar analyses using different databases may yield different results. We also recognise that the co-occurrence analysis and visualisation maps produced by the VOSviewer software have technical limitations.

## Figures and Tables

**Figure 1 ijerph-19-16427-f001:**
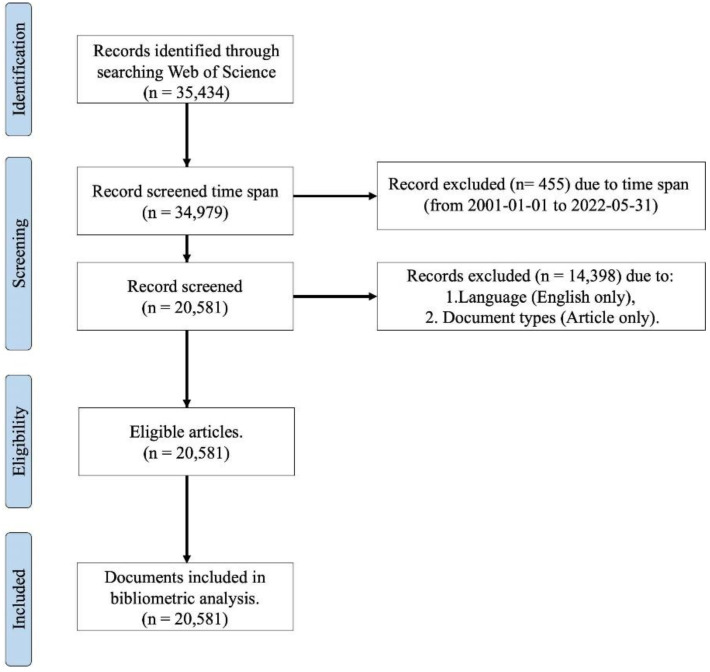
PRISMA flow diagram.

**Figure 2 ijerph-19-16427-f002:**
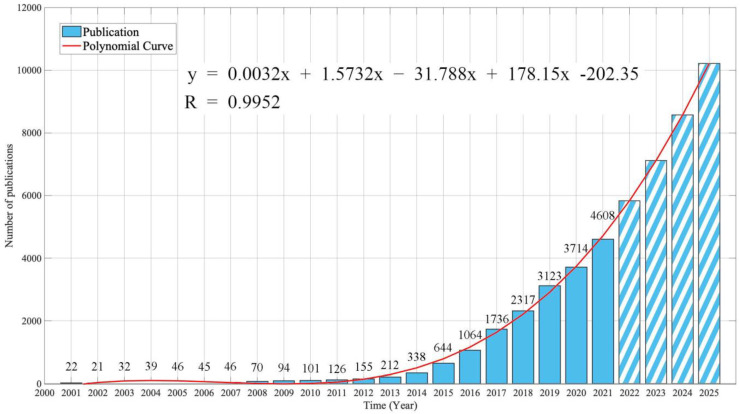
Temporal change of number of publications, 2001–2021. Solid bars are measured values and diagnal line bars (2022–2025) are estimated from a fourth polynomial regression.

**Figure 3 ijerph-19-16427-f003:**
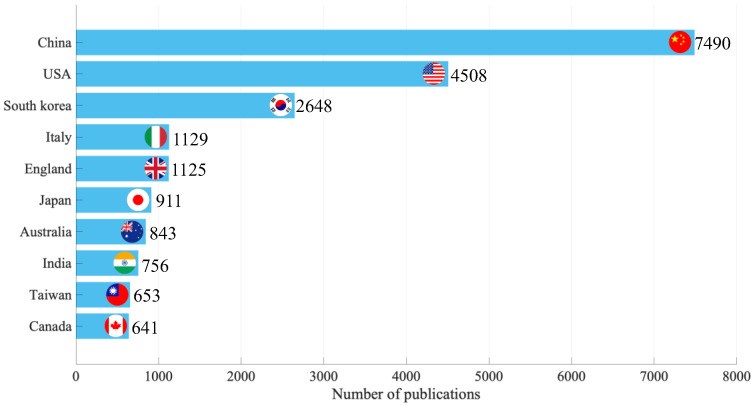
Comparison of number of publications by country, 2001–2022.

**Figure 4 ijerph-19-16427-f004:**
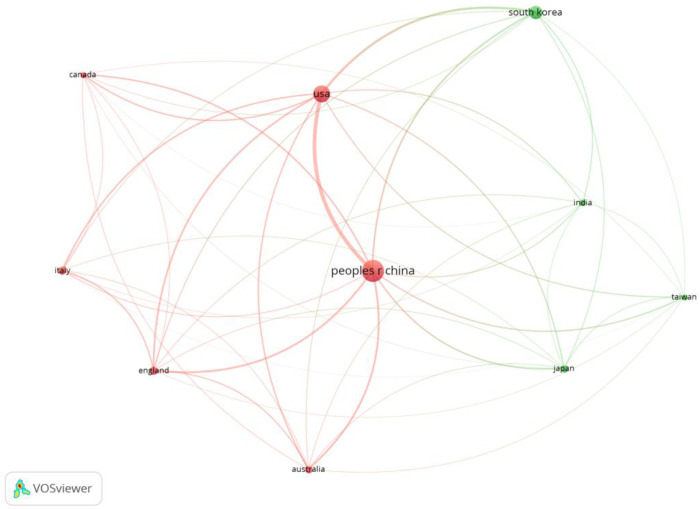
Co-authorship networks in the Top 10 countries with the largest number of papers, 2001–2022.

**Figure 5 ijerph-19-16427-f005:**
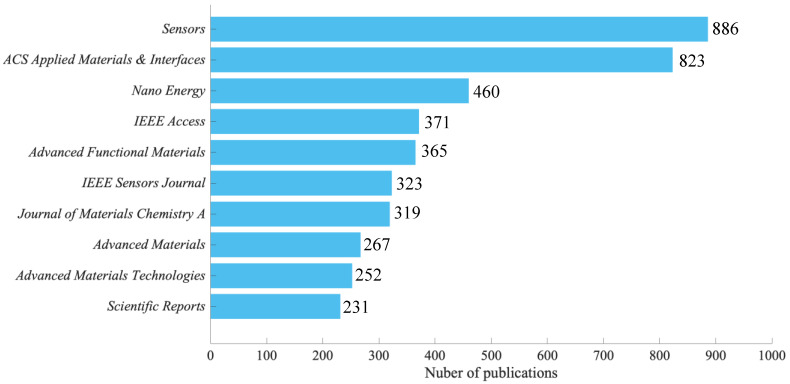
Top 10 journals publishing research on wearable device, 2001–2022.

**Figure 6 ijerph-19-16427-f006:**
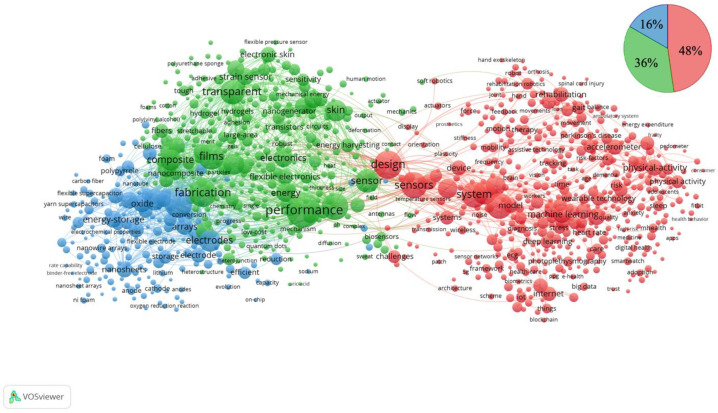
The co-occurrence network of the top 1000 keywords in wearable device research, 2001–2022.

**Figure 7 ijerph-19-16427-f007:**
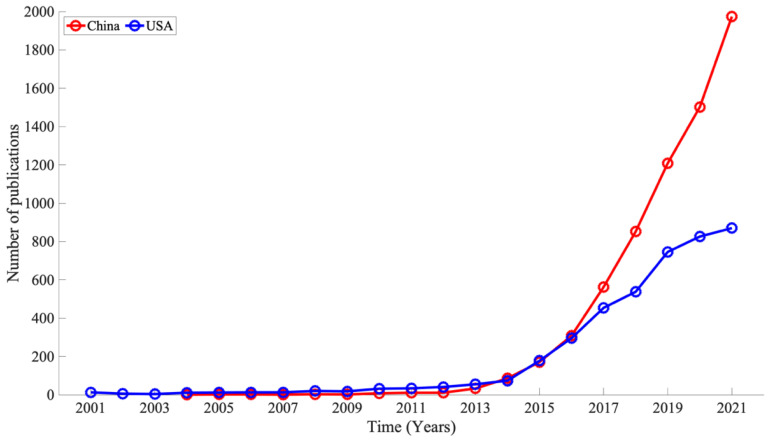
Comparison of temporal changes in the number of papers published in the China and USA, 2001–2021. No data exist for China from 2001 to 2003, as no relevant papers were published.

**Figure 8 ijerph-19-16427-f008:**
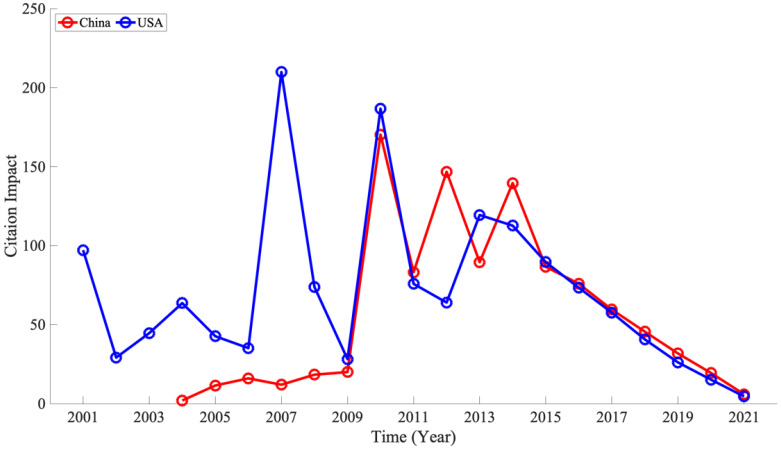
Comparison of citation impact in the Peoples r china and USA, 2001–2021. No data exist for China from 2001 to 2003, as no relevant papers were published.

**Figure 9 ijerph-19-16427-f009:**
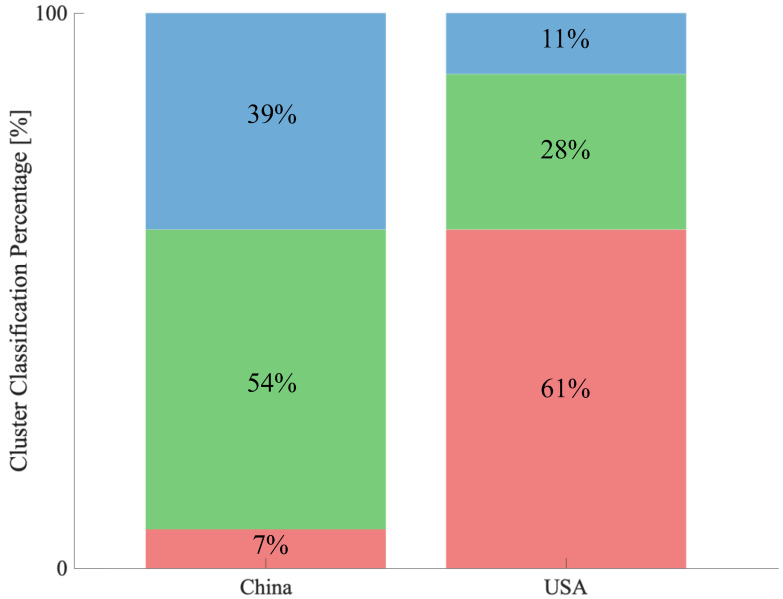
Cluster distribution map of author keywords used in China and USA, 2001–2021. The top 100 keywords with the highest number of co-occurrences among the author keywords are used (Red: cluster 1, Green: cluster 2, Blue: cluster 3).

**Table 1 ijerph-19-16427-t001:** Top 30 co-occurrence keywords by cluster, 2001–2022.

Rank	Cluster 1		Cluster 2		Cluster 3	
	Keyword	Weight	Keyword	Weight	Keyword	Weight
1	Design	849	Performance	889	Electrodes	617
2	Sensors	815	Devices	794	Graphene	608
3	Wearable devices	798	Sensor	754	High-performance	536
4	Wearable device	754	Fabrication	653	Flexible	478
5	System	736	Films	589	Oxide	457
6	Wearable	728	Skin	587	Arrays	452
7	Wearable sensors	622	Energy	583	Electrode	442
8	Behavior	607	Wearable electronics	577	Supercapacitors	435
9	Device	595	Pressure	556	Paper	426
10	Health	515	Electronics	546	Carbon	404
11	Systems	512	Transparent	541	Reduction	404
12	Model	501	Composite	540	Fiber	402
13	Technology	497	Carbon nanotubes	520	Reduced graphene oxide	398
14	Wearables	469	Nanoparticles	514	Nanowires	392
15	Machine learning	468	Flexible electronics	493	Supercapacitor	385
16	Challenges	463	Composites	492	Energy-storage	384
17	Wearable sensor	445	Stability	488	Nanosheets	373
18	Reliability	443	Temperature	482	Efficient	372
19	Monitoring	431	Film	478	Textiles	365
20	Management	397	Networks	474	Nanotubes	362
21	Wearable technology	397	Nanocomposites	469	Storage	358
22	Classification	394	Network	439	Polyaniline	349
23	Waling	394	Stain sensor	431	Nanostructures	342
24	Rehabilitation	377	Stain	429	Batteries	322
25	Physical-activity	376	Polymer	423	Flexibility	312
26	Recognition	373	Graphene oxide	420	Hybrid	305
27	Biomedical monitoring	369	Conductivity	404	Polypyrrole	302
28	Disease	362	Stain sensors	404	Conversion	297
29	Internet	361	Fibers	402	Energy storage	294
30	Accelerometer	360	Triboelectric nanogenerator	400	Capacitance	293

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
