# Peer review of "A Bibliometric Analysis of Wearable Device Research Trends 2001–2022—A Study on the Reversal of Number of Publications and Research Trends in China and the USA"

_ijerph, 2022, doi:10.3390/ijerph192416427_

Round 1
Reviewer 1 Report
The manuscript entitled “A bibliometric analysis of wearable device research trends 2001- 2 2022 -A Study on the Reversal of Number of Publications and Re- 3 search Trends in China and the USA” has been organized and developed in good shape. The results and findings are interesting. However, having reviewed the submitted manuscript, I addressed the following comments.

Author Response
November 25, 2022
Dear Reviewer,
First of all, we’d like to express our gratitude to you. Your comments and suggestions are invaluable for us to improve our paper.
We believe that we did our best in improvements according to all of your comments and suggestions as below, and we hope that our responses meet your expectations and your intentions.
We highly appreciate your cooperation.
Warm Regards,
Kota Kodama, PhD
Graduate School of Technology Management, Ritsumeikan University
2-150, Iwakura-cho, Ibaraki, Osaka, 567-8570, Japan
+81-72-665-2448
kkodama@fc.ritsumei.ac.jp
|
ID |
Comments and Suggestions |
Response |
|
Reviewer 1-1 |
The English writing level of the manuscript should be improved and punctuation error should be sorted out. |
Thank your suggestions regarding English proficiency. I have already used the editing services before the submission. However, it may not have been sufficient. Therefore, we would like to use the editing service once again during the final proofreading stage. |
|
Reviewer 1-2 |
Keywords: They should be in the alphabet order. In this paper: Bibliometric Analysis; Cluster Analysis; Co-occurrence Analysis; Wearable Device. |
Thank you very much for your advice. As suggested, we have modified the description in the Keywords as follows.
Keywords: Bibliometric Analysis; Cluster Analysis; Co-occurrence Analysis; Wearable Device |
|
Reviewer 1-3 |
Introduction: this section should be improved to show the advantages/disadvantages of references. In addition, it is highly recommended and strongly suggested that the authors add the following recently published paper to support the context of review of the wearable devices where it not highlighted it this section. https://doi.org/10.1109/TNSRE.2021.3136088 |
Thank you very much for your advice. Based on the reviewer's advice, we have modified the Introduction as follows.
Alternatively, a bibliometric analysis of wearable device research has been reported using wearable technology and other related products (e.g., smartwatches and smart glasses) as keywords [32]. However, the use of specific keywords narrows the research domain of interest and, trend analyses are limited to qualitative reviews of research papers under particular conditions, and none of them covers the wearable device research as whole. The reason for this is thought to be that this research domain covers many different fields, making it difficult to see the research domain as a whole. Thus, previous studies intentionally narrow the target research domain by selecting the in-tended use and industry or assigning specific keywords. Therefore, this study does cover the entire wearable device.
And, we checked the paper you suggested. It seems to fit as an example of prior bibliometric analysis research. Therefore, we have added it as reference [36] in the "2.1 Bibliometric analysis" section, line 82. |
|
Reviewer 1-4 |
Section 2. Method: It is highly recommended that authors use the PRISMA guidelines to show the results strategy clearly. This strategy is presented in Fig. 7 of the following paper: https://doi.org/10.1109/TNSRE.2021.3136088 |
Thank you very much for your advice. Based on the reviewer's advice, a new Figure 1 was established, showing PRISMA flow diagram. |
|
Reviewer 1-5 |
Section 3. Results: It is recommended that authors predict how many papers can be published in the next five years in this area. Please look at Fig. 16 of the paper above. |
Thank you for your suggestions to improve this paper. Based on the your advice, Figure 2 (previously Figure 1) has been revised. In addition, we added the following description to the 207 line of the “3.1 Characteristics of Research Domains “ section.
From the results of year 2001 through 2021, a fourth polynomial regression analysis was performed to estimate the number of publications to the year of 2025 (y=0.0032x^4+1.5732x^3-31.788x^2+178.15x^ -202.35, R^2=0.9952). As a result, the number of papers related to the area of research on wearable devices is expected to exceed 10,000 in 2025. |
|
Reviewer 1-6 |
Section 4. Discussion: This session is one of the most important parts of the paper and should be organized in a better shape in several paragraphs supporting each other. |
Thank your suggestions. Based on the reviewer's advice, we revised "4. Discussion" section structure with adding paragraphs. |
|
Reviewer 1-7 |
Section 5. Conclusion: It should be rewritten, where it is not written in a professional shape. Please look at the paper above for an example. |
Thank you for your suggestions. Based on the reviewer's advice, revised "5. Conclusions" section. |
|
Reviewer 1-8 |
Appendix 1: No table is available, and it is disappeared. |
I am sorry. We will check it thoroughly when reposting. |
Reviewer 2 Report
This manuscript conducts a bibliometric analysis of articles on wearable devices from 2001 to 2022 for mapping the intellectual structure in the research domain. Through various analyses of the collected articles, the authors clarify three clusters on two domains in balance — applied research domains and basic research domains, and compare the research emphasis in China and the USA. The manuscript is comprehensive. I have some questions and suggestions for this study.
1. In the analysis of citation impact, since the citation number of literature in different years is also affected by the year of publication, how do the authors identify the phenomenon of “contrary to the rapid increase in the number of publications in China” on page 8. The authors should clearly show their discussion here.
2. In the manuscript, three clusters are identified in the research domain of wearable devices. I am curious about the classification basis of the three clusters, as some similar keywords listed in different clusters, such as “Device” and “Devices” are listed in Cluster 1 and 2, respectively. Could this basis be discussed further?
3. The authors claim that the research inclinations in China and the USA are different. Can such differences be analyzed from the perspectives of time? In other words, do the research inclinations in both countries show temporal changes?
4. On page 10, as mentioned “In this research domain, where both basic and applied research are important, the difference in direction between China and the USA is a desirable outcome”, could the authors provide more discussion about the “desirable outcome”? How do the authors view the future direction of such difference? Will this difference change in the future? The authors may discuss more the outlook in the future trends.
5. Numerous formatting issues and language issues should be addressed (examples below).
1) “2. Method” on page 2 is “2. Methods”.
2) The caption of Figure 2 is out of place.
3) The references should be reformatted according to the journal standard.
Author Response
November 25, 2022
Dear Reviewer,
First of all, we’d like to express our gratitude to you. Your comments and suggestions are invaluable for us to improve our paper.
We believe that we did our best in improvements according to all of your comments and suggestions as below, and we hope that our responses meet your expectations and your intentions.
We highly appreciate your cooperation.
Warm Regards,
Kota Kodama, PhD
Graduate School of Technology Management, Ritsumeikan University
2-150, Iwakura-cho, Ibaraki, Osaka, 567-8570, Japan
+81-72-665-2448
kkodama@fc.ritsumei.ac.jp
|
ID |
Comments and Suggestions |
Response |
|
Reviewer 2-1 |
In the analysis of citation impact, since the citation number of literature in different years is also affected by the year of publication, how do the authors identify the phenomenon of “contrary to the rapid increase in the number of publications in China” on page 8. The authors should clearly show their discussion here. |
Thank you very much for your advice. It was a difficult sentence to understand as you said. Therefore, we added the following description to the 349 line of the “ 3.3 Comparing two countries' contributions to the research domains (China vs. USA) “ section.
Although an effect of the calculation based on single year to capture recent increase and a time peri-od after the appearance of a publication (often referred as citation window) should be considered [52], of course, recent trend of citation impact suggests qualitative difference of wearable device re-search between China and USA. |
|
Reviewer 2-2 |
In the manuscript, three clusters are identified in the research domain of wearable devices. I am curious about the classification basis of the three clusters, as some similar keywords listed in different clusters, such as “Device” and “Devices” are listed in Cluster 1 and 2, respectively. Could this basis be discussed further? |
Thank you very much for your advice. As you mentioned, our explanation was inadequate. Therefore, we added the following description to the 133 line of the “2.3 Co-occurrence analysis of keywords “ section.
This study did not take steps to merge singular and plural keywords (e.g., singular "devices", "sensors" and plural "devices", "sensors", etc.). In this study, what each cluster means was judged by keywords except above one. The reason is that these keywords, which are often listed by authors in wearable device research field, can generate false co-occurrence even if there is not actual relationship. Therefore, these keywords (i.e., “devices” and “devices”) exist as separate nodes and are each classified into a cluster.
In addition, we added the following description to the 480 line of the “5. Conclusions “ section as an issue.
We recognize the limitations in this study. In the cluster classification, we did not unify some author keywords for simplicity because we could judge meanings of each cluster without lemmatization. To conduct more precise analysis, however, preprocessing of author keywords, such as merging singular and plural keywords or setting some stopwords will be required. Furthermore, we did not analyze trends in this research area in terms of time. This is a problem with the tool used (VOSviewer is not suitable for analyzing from a time perspective. CiteSpace would be a better choice). |
|
Reviewer 2-3 |
The authors claim that the research inclinations in China and the USA are different. Can such differences be analyzed from the perspectives of time? In other words, do the research inclinations in both countries show temporal changes? |
Thank you for reading the details and your suggestions to improve this paper. However, this study has not been analyzed from a time perspective because of the tools used. However, we think it is an interesting point. Therefore, we have added a description of this issue in "5. Conclusion" section (see Reviewer 2-2).
|
|
Reviewer 2-4 |
On page 10, as mentioned “In this research domain, where both basic and applied research are important, the difference in direction between China and the USA is a desirable outcome”, could the authors provide more discussion about the “desirable outcome”? How do the authors view the future direction of such difference? Will this difference change in the future? The authors may discuss more the outlook in the future trends. |
Thank you very much for your advice. Our explanation was inadequate. Therefore, we added the following description to the 473 line of the “5. Conclusions “ section.
… (e.g., recent smart watches can measure body temperature, blood flow, blood pressure, and blood oxygen levels, but research is being conducted to measure blood sugar levels [58] ). Thus, technological developments and breakthroughs from basic research can create new uses (applied research), and new applied research can motivate new basic research. And it will open up new markets. In this research area, where both fundamentals and applications are important, the different directions of China and the USA are a desirable outcome in a mutually supportive manner.
In addition, we added the following description to the 439 line of the “4. Discussion “ section as an issue.
In other words, typically through the application in the USA, outcomes derived from the use case of wearable devices in the real world can accelerate active knowledge transfer into the research field for innovative component to improve the wearable device performances, where China has superiority, vice versa. |
|
Reviewer 2-5 |
Numerous formatting issues and language issues should be addressed (examples below). 1) “2. Method” on page 2 is “2. Methods”. 2) The caption of Figure 2 is out of place. 3) The references should be reformatted according to the journal standard. |
Thank you for reading the details and your suggestions to improve this paper. Based on the reviewer's advice, the description was modified. Moreover, we already use editing services before submitting our papers. But it may not have been sufficient. When this paper is accepted, we would like to use the editing service again during the final proofreading stage. |
Round 2
Reviewer 1 Report
The paper is improved significantly, and it is highly recommended for publication.
Reviewer 2 Report
The authors have done a great job addressing my comments.